# Screening of Bacterial Endophytes Able to Promote Plant Growth and Increase Salinity Tolerance

**Elisa Gamalero** [1], **Nicoletta Favale** [2], **Elisa Bona** [3,*], **Giorgia Novello** [1], **Patrizia Cesaro** [1], **Nadia Massa** [1], **Bernard R. Glick** [4], **Ma del Carmen Orozco-Mosqueda** [4], **Graziella Berta** [1] and **Guido Lingua** [1]

1. Dipartimento di Scienze e Innovazione Tecnologica, Università del Piemonte Orientale, Viale T. Michel 11, 15121 Alessandria, Italy; elisa.gamalero@uniupo.it (E.G.); giorgia.novello@uniupo.it (G.N.); patrizia.cesaro@uniupo.it (P.C.); nadia.massa@uniupo.it (N.M.); graziella.berta@uniupo.it (G.B.); guido.lingua@uniupo.it (G.L.)
2. Department of Life Sciences and Biotechnologies, University of Ferrara UNIFE, 44121 Ferrara, Italy; nicoletta.favale@unipr.it
3. Dipartimento di Scienze e Innovazione Tecnologica, Università del Piemonte Orientale, Piazza San Eusebio 5, 13100 Vercelli, Italy
4. Department of Biology, University of Waterloo, Waterloo, ON N2L 3G1, Canada; glick@uwaterloo.ca (B.R.G.); gsantoyo@umich.mx (M.d.C.O.-M.)
* Correspondence: elisa.bona@uniupo.it

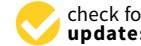

**Featured Application: These five strains were characterized in order to use them to promote plant growth in the presence of salt stress.**

**Abstract:** Bacterial endophytes can colonize plant tissues without harming the plant. Instead, they are often able to increase plant growth and tolerance to environmental stresses. In this work, new strains of bacterial endophytes were isolated from three economically important crop plants (sorghum, cucumber and tomato) grown in three different regions in soils with different management. All bacterial strains were identified by 16S rRNA sequencing and characterized for plant beneficial traits. Based on physiological activities, we selected eight strains that were further tested for their antibiotic resistance profile and for the ability to efficiently colonize the interior of sorghum plants. According to the results of the re-inoculation test, five strains were used to inoculate sorghum seeds. Then, plant growth promotion activity was assessed on sorghum plants exposed to salinity stress. Only two bacterial endophytes increased plant biomass, but three of them delayed or reduced plant salinity stress symptoms. These five strains were then characterized for the ability to produce the enzyme 1-aminocyclopropane-1-carboxylate (ACC) deaminase, which is involved in the increase of stress tolerance. *Pseudomonas brassicacearum* SVB6R1 was the only strain that was able to produce this enzyme, suggesting that ACC deaminase is not the only physiological trait involved in conferring plant tolerance to salt stress in these bacterial strains.

**Keywords:** endophyte isolation; physiological activities; plant growth promotion; salt stress; ACC deaminase

## 1. Introduction

Bacteria living in the rhizosphere can be classified based on their degree of intimacy with the root: bacteria colonizing the root surface are defined as rhizospheric, while others, such as endophytes, enter inside plant tissues and are often found in roots or even in shoots [1].

According to one of the first definitions, endophytes are bacteria or fungi that are able to live in the plant endosphere during all or part of their life cycle without inducing any damage to the plant [1]. Later, Hallman et al. [2] suggested to classify endophytes as any bacterium that can be isolated from surface-sterilized plant tissue that does not harm the plant. More recently, Hardoim et al. [3] suggested that endophytes are bacteria, archaea, fungi, and protists able to colonize a plant's interior, regardless of their interaction with the plant.

Except for already established seed-endophytes, the most common mode of entry of endophytic bacteria into plant tissues is through primary and lateral root cracks, and diverse tissue wounds that release plant metabolites which attract bacterial cells [4]. Less frequently, bacterial endophytes enter plants through stomata (particularly on leaves and young stems), lenticels, germinating radicles and emerging lateral roots or root hair cells [4]. The soil is a sink of rhizospheric bacteria that are able to efficiently colonize root surfaces and penetrate inside plant tissues [4].

Although many of the genes involved in endophytic behavior have not yet been identified, the occurrence of genetic differences between bacteria living in the rhizosphere and those able to colonize the interior of plant organs has been speculated [5]. Living inside the plant offers the possibility of establishing more efficient plant–bacterial communication [6,7]. Moreover, endophytic bacteria are less exposed to variations in soil properties such as pH, water availability, and competition with the external microbiota [4]. From the plant interior, bacterial endophytes can improve plant growth and health by several mechanisms including the solubilization of phosphate [8], synthesis of phytohormones [9,10], nitrogen fixation [11–13], siderophore production [14,15], and suppression of phytopathogenic microorganisms [16,17].

In addition, bacterial endophytes can increase plant tolerance to stress through the reduction of ethylene, which is synthesized by a plant exposed to either biotic stresses (phytopathogenic fungi, bacteria and nematodes) or abiotic stresses (higher concentration of heavy metals, organic pollution, flooding, drought and salinization). This occurs through the action of the enzyme 1-aminocyclopropane-1-carboxylate (ACC) deaminase, which cleaves ACC (the immediate precursor of ethylene) to alpha-ketobutyrate and ammonia. As a direct consequence, the amount of ethylene in the host plant is reduced [18,19].

Soil salinization is one of the most common environmental factors limiting the yield and health of crop plants [20]. Nearly 831 Mha of land worldwide suffers from an excess of salt [20], while salinity related to human activities affects about 76 Mha of land worldwide (Food and Agriculture Organization of the United Nations- FAO 2015) [20]. In Europe, Mediterranean countries are more affected by salinization, which is often coupled with soil alkalization; in Italy, soil salinization accounts for 3.2 Mha [20]. Plants respond to drought and salinity in a similar way, which is related to water stress. High soil salinity leads to osmotic stress, $Na^+$ and $Cl^-$ toxicity, ethylene production, plasmolysis, nutrient imbalance, the production of reactive oxygen species (ROS), and interference with the photosynthetic process. These adverse effects are induced in plants growing in these conditions, and both growth and yield are impaired. As a consequence, seed germination is inhibited, seedling growth and vigor are reduced, and flowering and fruit yield are decreased.

Based on the need to address the problems caused by soil salinization, this work aimed to isolate new bacterial endophytes from soils that were managed in different ways (high mountain pasture, kitchen garden and woody soils), using three agriculturally important plants (sorghum, cucumber and tomato) grown on these soils, looking at possible beneficial physiological traits induced by the endophytes in plants under normal or salt-stressed conditions. In particular, the following traits of bacterial endophytes were analyzed in this study: siderophore production, phosphate solubilization, synthesis of auxin, capacity to recolonize plant tissues, capacity to promote plant growth, resistance to antibiotics, ACC deaminase synthesis, and the ability to increase plant tolerance to salinity stress.

## 2. Materials and Methods

### *2.1. Sampling Site*

In order to isolate bacterial endophytes, three soil samples were collected from three different areas in Italy: (1) a high mountain pasture (1500 a.s.l., located in Capanne di Cosola, Val Borbera, Alessandria); (2) a kitchen garden located in Gavi (Alessandria); and (3) a chestnut wood located close to Lago d'Orta (Orta San Giulio, Novara) (Figure 1). Climatic data (daily precipitation and temperature) for each sampling site are reported in Figure 2.

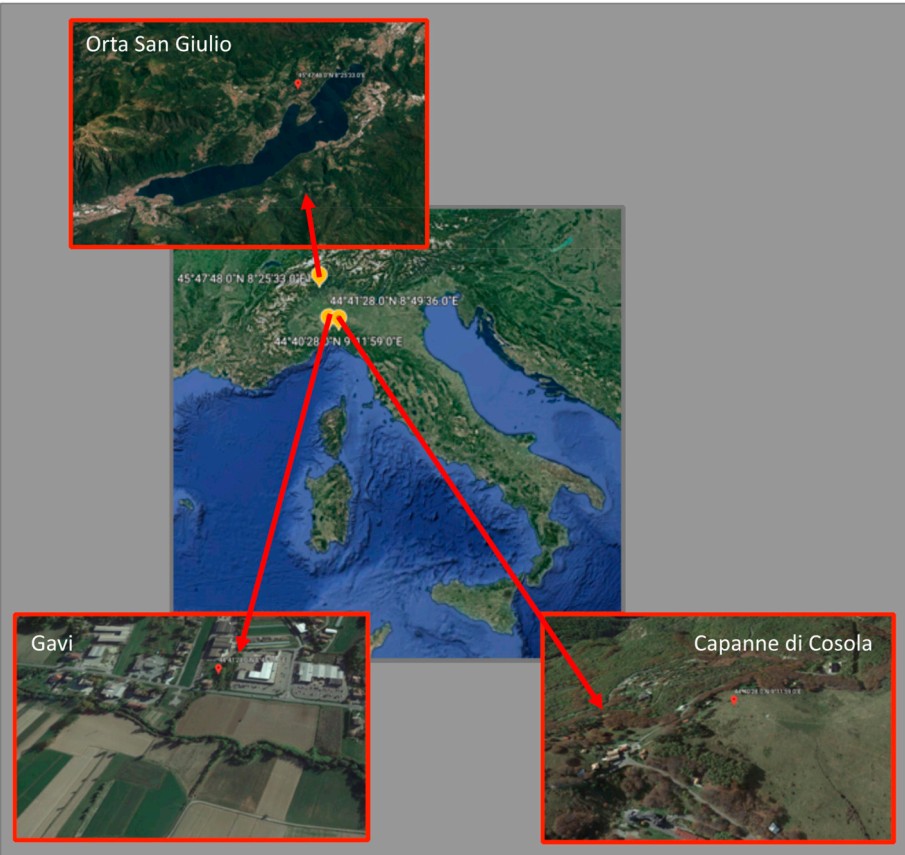

**Figure 1.** Areas where soil has been collected: a high mountain pasture (1500 a.s.l. located in Capanne di Cosola, Val Borbera, Alessandria, Italy—VB, 44°40′20″ N–9°11′59″ E), a kitchen garden located in Gavi (Alessandria, Italy—O, 44°41′28″ N–8°49′36″ E) and a chestnut wood located close to Lago d'Orta (Orta San Giulio, Novara, Italy—L, 45°47′48″ N–8°25′33″ E).

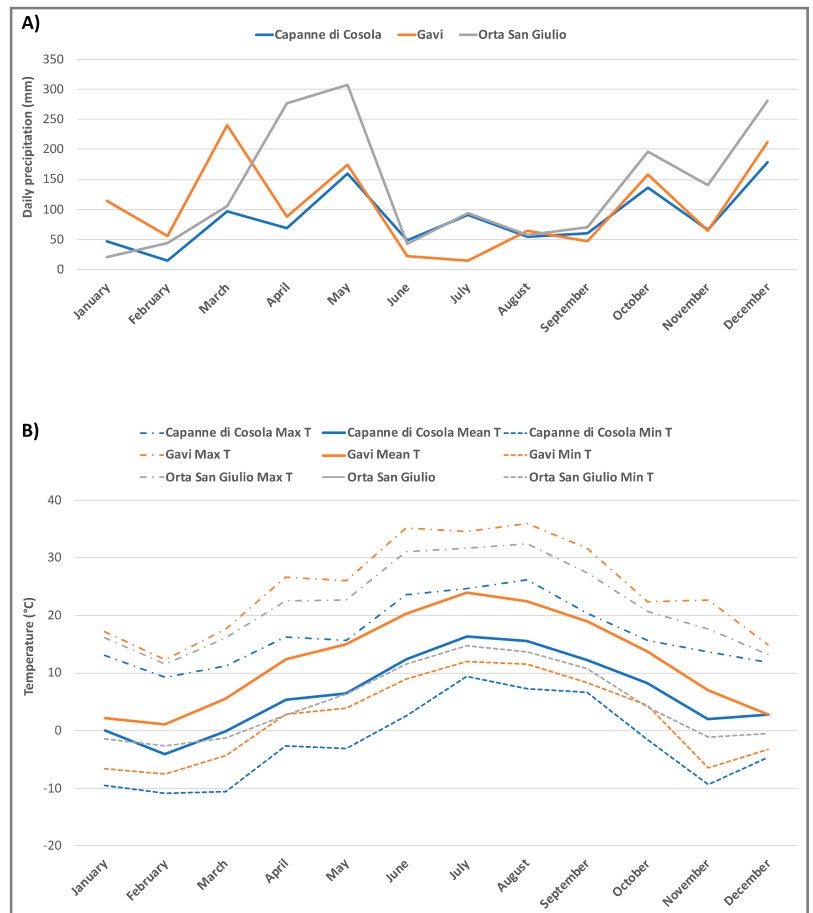

**Figure 2.** Climatic data for the three sampling sites (Capanne di Cosola, Gavi, Orta San Giulio). (**A**) Daily mean precipitation (mm); (**B**) temperature (T) (mean temperature, minimum temperature and maximum temperature) (°C).

### 2.2. Seed Sterilization

About 100 cucumber (*Cucumis sativus*—C) (Ortaggi ABF, Italy), sorghum (*Sorghum bicolor*—S) (Fratelli Ingegnoli spa) and tomato seeds (*Solanum lycopersicum*—P) (Ortaggi ABF) were surface sterilized with NaClO diluted 1:5 in sterile deionized water for 5 min. Then, seeds were washed once with 1% Tween for 5 min, 6 times with deionized water for 5 min and twice with deionized water for 20 min. Seeds were then incubated at 25 °C for 72 h.

In order to exclude the presence of exogenous microorganisms, surface sterilization of seeds was assessed by incubating some seeds on trypticase soy agar (TSA; Biolife, Milan, Italy) at 28 °C for 96 h [21]. One germinated seed of each plant species was sown in 500 mL plastic pots previously sterilized in 20% NaClO containing 400 mL of each soil type (7 seeds × 3 soil types × 3 plant species, for a total of 63 plants). Plants were grown under greenhouse conditions, watered three times per week with tap water (on Monday, Wednesday, and Friday) and harvested after 1 month.

### 2.3. Isolation of Bacterial Endophytes

The isolation of bacterial endophytes was performed following the method described by Rashid et al. [22] and modified as follows: each harvested plant was divided into roots (R), stems (F), and leaves (L). The three plant samples were washed with tap water in order to remove adherent soil, gently dried by paper and weighed. Surface disinfection of the plant tissues was carried out by a 5 min treatment with NaClO (5.25% available chlorine), followed by three treatments with 3% hydrogen peroxide solution for 5 min for each treatment. Plant tissues were then washed 6 times in sterilized

deionized water; for the first rinse, a solution of 10% Tween 20 was added. Plant tissues were then placed on TSA with cycloheximide (100 µg mL$^{-1}$) in order to assess surface sterility. Plant tissues were then homogenized in 10 mL of 3× Ringer's solution [23] using a mortar and pestle, previously sterilized at 121 °C for 15 min, and incubated at room temperature (21–23 °C) in an orbital shaker for 1 h. Suspensions were serially diluted to $1 \times 10^{-3}$ in 3× Ringer's solution. Aliquots (100 µL) of each dilution were spread in triplicate onto TSA with cycloheximide. The plates were incubated at 28 °C for 72 h. Morphologically different bacterial colonies were selected, isolated and maintained in glycerol culture stocks at −80 °C.

## 2.4. Bacterial Endophyte Identification

Bacterial strain identification was performed by 16S rDNA sequencing [24]. Genomic DNA was extracted using a NucleoSpin tissue DNA purification kit (Macherey-Nagel, M-Medical, Cornaredo, Milan, Italy) following the manufacturer's instructions. PCR amplification of 16S ribosomal DNA (rDNA) was performed using the modified universal bacterial primers fD1 (5′-AGAGTTTGATCCTGGCTCAG-3′) and RP2 (5′-ACGGCTACCTTGTTACGACTT-3′) [25]. The PCR cycle included the following steps: initial denaturation at 94 °C for 5 min; 34 cycles at 94 °C for 1 min, 60 °C for 1 min, and 72 °C for 2 min 30 s; and 72 °C for 10 min as the final extension time. Both negative (PCR mixture without DNA template) and positive (PCR with a positive DNA template) controls were included for each PCR reaction. The amplification products were visualized on an 1.2% agarose gel containing ethidium bromide using the Chemidoc system (Biorad). Purification of the amplified DNA fragments were performed with the NucleoSpin Extract II kit (Macherey-Nagel, M-Medical, Cornaredo, Milan, Italy) and sequenced by BMR Genomics (Padua, Italy). Electropherograms were analyzed using Finch TV version 4.0 (Applied Biosystems, Milan, Italy) and transformed in a text file for further bioinformatic analyses. DNA sequences were blasted against the bacterial 16S rDNA reference sequences of the National Center for Biotechnology Information (NCBI) database [24].

## 2.5. Characterization of Physiological Plant-Beneficial Activities

All strains were characterized for their physiological activities that could be involved in plant growth promotion. Siderophore production was evaluated on universal Chrome Azurol S (CAS) agar [26]. Each bacterial strain was inoculated at the center of the plate. The appearance of an orange halo around the colony, after incubation at 28 °C for 3 days, indicated siderophore synthesis. All measurements were done in triplicate. The ratio between the two diameters of the halo and the two diameters of the colony was then calculated.

Phosphate solubilization was assayed on two different media as described by Bona and co-workers [24], one containing dicalcium phosphate (DCP) ($NH_4Cl$, 4.25 g L$^{-1}$; NaCl, 0.85 g L$^{-1}$; $MgSO_4 \cdot 7H_2O$, 0.85 g L$^{-1}$; glucose, 8.5 g L$^{-1}$; $K_2HPO_4$, 2 g L$^{-1}$; $CaCl_2 \cdot 2H_2O$, 4 g L$^{-1}$; and agar, 17 g L$^{-1}$) and one containing tricalcium phosphate (TCP) ($NH_4Cl$, 5 g L$^{-1}$; NaCl, 1 g L$^{-1}$; $MgSO_4 \cdot 7H_2O$, 1 g L$^{-1}$; glucose, 10 g L$^{-1}$; $Ca_3(PO_4)_2$, 40 g L$^{-1}$; and agar, 20 g L$^{-1}$). A colony of each strain was inoculated at the center of the plate and incubated. After incubation at 28 °C for 15 days, DCP solubilization was indicated by a clarification halo around the colony, while the capability to solubilize TCP was identified by bacterial growth on the medium.

Indole-3-acetic acid (IAA) synthesis was quantified following the method described by De Brito Alvarez et al. [27]. The bacterial strains were inoculated onto a nitrocellulose disk placed on 10% TSA with 5 mM L-tryptophan and incubated at 28 °C for 3 days. The membranes were dipped in 20 mL of Salkowsky's reagent (2% $FeCl_3$ in 35% perchloric acid); the development of a red/pink halo around the colony was indicative of IAA production.

## 2.6. Antibiotic Resistance Profile

In order to assess antibiotic resistance, bacterial suspensions with an optical density of 0.5 at λ 600 nm (corresponding to about $1 \times 10^8$ colony forming units (CFU)/mL determined using both

the optical density measurement and the dilution plating methods) were obtained in 0.1 M MgSO$_4$. One hundred microliters of each bacterial suspension was spread on the surface of a Petri dish containing Mueller Hinton agar (Biolife, Milan, Italy). Disks containing both wide spectrum antibiotics and antibiotics specific for Gram negative bacteria, such as ceftazimide (30 μg), cotrimoxazole (25 μg), gentamycin (10 μg), ciprofloxacin (5 μg), nalidixic acid (30 μg), nitrofurans (100 μg), cefoperazone (30 μg), phosphomycin (50 μg), cefixime (10 μg), and norfloxacin (10 μg), were deposited on the agar and the plates were incubated at 28 °C for 24 h. The diameter of the growth inhibition halo was measured and then compared to the value reported by European Committee on Antimicrobial Susceptibility Testing (EUCAST) and in order to define each bacterial strain as resistant (R), sensitive (S) or intermediate (I).

### 2.7. Determining Capability to Colonize Plant Tissue and Promote Plant Growth

The ability of eight selected bacterial strains to colonize sorghum plant tissues and promote plant growth was assessed. A spontaneous mutant resistant to 100 μg/mL rifampicin was obtained for each strain. Rifampicin resistance is a marker gene that allows us to distinguish the inoculated strain from other possible environmental microorganisms that may contaminate the pots. Resistance was induced by inoculating the bacterial strain on TSA medium with an increasing concentration of rifampicin. Seeds were sterilized as previously described and incubated for 3 days at 25 °C. Germinated seeds were then dipped for 20 min in a bacterial suspension containing $1 \times 10^8$ CFU/mL in 0.1 M MgSO$_4$ and sown in sterilized 500 mL plastic pots containing 400 mL of sterile quartz sand. Each bacterial strain was used to inoculate 10 seeds of each plant species from which it originated. Ten seeds per plant species did not receive any bacterial inoculation and were used as negative controls. A total of 90 plants (10 plants per 8 bacterial strains + 10 controls) were cultivated in a growth chamber with a 16/8 h light/dark photoperiod, at a light/dark temperature of 24/20 °C, at 150 μE m$^{-2}$ s$^{-1}$ light irradiance at pot height during the light photoperiod at 60% relative humidity, and were watered to saturation three times per week with modified Long Ashton growth medium. Plant growth parameters such as plant fresh weight, root and shoot fresh weight, and number of leaves were evaluated at harvest.

### 2.8. Evaluation of ACC Deaminase Activity

Five out of the selected eight strains were able to colonize internal plant tissues, and they were tested for the capability to synthesize ACC deaminase as described by Penrose and Glick [28] and compared with a standard curve of α-ketobutyrate ranging from 0.05 to 0.5 μmol.

*Pseudomonas* sp. UW4 [29] (able to synthesize ACC deaminase and produce α-ketobutyrate) and its mutant (unable to synthesize ACC deaminase or produce α-ketobutyrate) were used as positive and negative controls, respectively.

### 2.9. Determination of Minimal Inhibitory Concentration for the Growth of Bacterial Strains in the Presence of Salt Stress

The effect of salt (sodium chloride) on the growth of selected strains was investigated using a microdilution method.

A 24 h culture of each bacterial strain was diluted in MgSO$_4$ (0.1 M) buffer to a final concentration of $1 \times 10^6$ CFU/mL. Sodium chloride (NaCl, Fluka) was dissolved in tryptic soy broth (TSB, Sigma) to obtain four concentrations (28.8%, 24.8%, 21.8% and 18.8%). Serial dilutions (1:2) of these NaCl concentrations were prepared in a 96-well microtiter plate. Each dilution was then inoculated with 100 μL of the cell suspension. A negative control, containing TSB with each of the NaCl dilutions, and a positive control containing TSB with bacterial strains at the same NaCl concentrations, were also performed. All plates were incubated at 28 °C for 48 h. The well was considered positive when the presence of a bacterial cell pellet was apparent. Each experiment was performed in triplicate.

### 2.10. Effects of Bacterial Endophytes on Plants Exposed to Salt Stress

The effects of the five bacterial endophytes able to synthesize ACC deaminase on sorghum plants exposed to salt stress was evaluated with this experiment. Seed sterilization, bacterial inoculation and plant growth conditions have been previously described. A total of 49 plants (7 plants inoculated with each of the five bacterial strains and exposed to salinity, plus 7 uninoculated plants not exposed to salinity, plus 7 uninoculated plants exposed to salinity) were prepared. During the first 7 days of growth, all plants received sterile Long Ashton growth medium three times a week. Treated plants were fed with sterile Long Ashton medium containing 150 mM NaCl (equal to 0.9% NaCl).

The appearance of symptoms of salt stress were monitored every 2 days and evaluated according to the following damage intensity scale: 0 = no symptoms, 1 = weak symptoms (yellowing of the leaf tips), 2 = strong symptoms (leaves dried or completely yellow), and 3 = plant death. After 20 days of salt stress exposure, surviving plants were harvested and plant, shoot and root fresh weight, and number of leaves were measured.

### 2.11. Statistical Analysis

According to Loper et al. [30], bacterial populations approximate an exponential normal distribution; therefore, values of bacterial density were logarithmically transformed before analysis. Data of the plant parameters were analyzed by ANOVA. Statistical analyses were performed with STATVIEW 4.5 (Abacus Concepts, Berkeley, CA, USA); data were compared by one-way ANOVA, followed by a *post hoc* Fisher's Protected Least Significant Difference (PLSD) test ($p \leq 0.05$).

## 3. Results

A total of 60 bacterial endophyte strains, belonging to 29 different species, were isolated from the three plant species grown on different soils: 19 from cucumber, 14 from sorghum and 27 from tomato. In particular, 60%, 16.7% and 23.3% of the bacterial endophytes were isolated from plants grown in pasture soil, in vegetable garden soil and in woody soil, respectively. The complete list of the bacterial strains containing their origin, taxonomic identification, GenBank accession number and qualitative evaluation of their plant beneficial physiological traits are reported in Table 1.

These endophytes were tested for their physiological activities involved in plant growth promotion, including auxin synthesis, solubilization of di- and tri-calcium phosphate and siderophore production (Table 1 and Figure 3). Sixteen out of the 60 bacterial isolates (26.7%) were able to synthesize IAA: 56.2% were from cucumber, 25% from sorghum and 18.7% from tomato. A total of 18 strains (30%) solubilized phosphate on DCP: 5.5% from cucumber, 38.9% from sorghum and 55.5% from tomato. None of the tested strains were able to solubilize tricalcium phosphate. Finally, 19 strains were positive for siderophore production (31.7%): 21% were from cucumber, 42.1% from sorghum and 36.8% from tomato (Table 1). The proportion of bacterial endophytes able to synthesize IAA, solubilize phosphate and produce siderophore was highest in pasture soil (55.5%, 76.2% and 57.1%, respectively).

**Table 1.** List of bacterial strains isolated from internal plant tissues and characterization of plant-beneficial physiological activities.

| Strain | Origin | Taxonomic Identification | GenBank Accession Number | IAA [a] (Intensity Colour Scale 0–5) | Siderophore (HD/CD [b] cm) | Phosphate Solubilisation (DCP [c], HD/CD cm) |
|---|---|---|---|---|---|---|
| CVB2R4 | cucumber root, pasture soil | *Herbaspirillum lusitanum* | KX436986 | 3 | 0.00 | 0.00 |
| CVB2R5 | cucumber root, pasture soil | *Herbaspirillum lusitanum* | KX436987 | 3 | 0.00 | 0.00 |
| CVB3S1 | cucumber shoot, pasture soil | *Acinetobacter johnsonii* | KX436989 | 0 | 0.00 | 0.00 |
| CVB3S2 | cucumber shoot, pasture soil | *Acinetobacter johnsonii* | KX437624 | 1 | 0.00 | 0.00 |
| CVB3S4 | cucumber shoot, pasture soil | *Acinetobacter johnsonii* | KX437625 | 0 | 0.00 | 0.00 |
| CVB3S5 | cucumber shoot, pasture soil | *Stenotrophomonas rhizophila* | KX437653 | 3 | 0.00 | 0.00 |
| CO3R3 | cucumber root, vegetable garden | *Agrobacterium tumefaciens* | KX429744 | 4 | 2.96 | 0.00 |
| CO3L1 | cucumber leaf, vegetable garden | *Agrobacterium tumefaciens* | KX429748 | 0 | 0.00 | 0.00 |
| CO4R1 | cucumber root, vegetable garden | *Agrobacterium tumefaciens* | KX429749 | 0 | 0.00 | 0.00 |
| CO4S1 | cucumber shoot, vegetable garden | *Rhizobium radiobacter* | MF993501 | 4 | 3.53 | 0.00 |
| CO4S3 | cucumber shoot, vegetable garden | *Agrobacterium tumefaciens* | KX436985 | 2 | 2.73 | 0.00 |
| CO5R1 | cucumber root, vegetable garden | *Micrococcus yunnanensis* | KX436988 | 0 | 0.00 | 0.00 |
| CL1S1 | cucumber shoot, chestnut woody soil | *Paenibacillus graminis* | KX404972 | 0 | 0.00 | 0.00 |
| CL1S2 | cucumber shoot, chestnut woody soil | *Bacillus pumilus* | KX421200 | 2 | 0.00 | 0.00 |
| CL6S1 | cucumber shoot, chestnut woody soil | *Bacillus cereus* | KX421202 | 0 | 0.00 | 0.00 |
| CL6S2 | cucumber shoot, chestnut woody soil | *Bacillus pumilus* | KX429743 | 2 | 0.00 | 1.72 |
| CL6S5 | cucumber shoot, chestnut woody soil | *Bacillus muralis* | KX429745 | 0 | 0.00 | 0.00 |
| CL6S9 | cucumber shoot, chestnut woody soil | *Bacillus muralis* | KX429746 | 0 | 3.44 | 0.00 |
| CL6S10 | cucumber shoot, chestnut woody soil | *Terribacillus goriensis* | KX429747 | 0 | 0.00 | 0.00 |
| SVB3R1 | sorghum root, pasture soil | *Bacillus cereus* | KX444208 | 0 | 0.00 | 0.00 |
| SVB3R2 | sorghum root, pasture soil | *Pseudomonas migulae* | MF993051 | 0 | 4.51 | 1.45 |
| SVB3R3 | sorghum root, pasture soil | *Pseudomonas migulae* | MF993115 | 0 | 5.63 | 0.62 |
| SVB3R4 | sorghum root, pasture soil | *Pseudomonas migulae* | MF993122 | 0 | 4.10 | 1.45 |
| SVB3R5 | sorghum root, pasture soil | *Pseudomonas* sp. | KX447593 | 0 | 3.93 | 1.72 |
| SVB6R1 | sorghum root, pasture soil | *Pseudomonas brassicacearum* | KX444672 | 3 | 5.55 | 2.92 |
| SVB6R2 | sorghum root, pasture soil | *Paenibacillus lautus* | KX444671 | 1 | 0.00 | 0.00 |
| SL3R1 | sorghum root, chestnut wood | *Brevibacterium frigoritolerans* | KX447586 | 0 | 2.78 | 0.00 |
| SL3R3 | sorghum root, chestnut wood | *Bacillus anthracis* | KX442610 | 0 | 0.00 | 0.00 |
| SL3R4 | sorghum root, chestnut wood | *Paenibacillus illinoisensis* | KX442643 | 0 | 0.00 | 1.52 |
| SL3R5 | sorghum root, chestnut wood | *Bacillus muralis* | KX444201 | 0 | 2.97 | 0.00 |
| SL3R6 | sorghum root, chestnut wood | *Pseudomonas* sp. | MF993023 | 2 | 2.76 | 1.50 |
| SL3R8 | sorghum root, chestnut wood | Bacillaceae bacterium | KX447594 | 0 | 0.00 | 0.00 |
| SL3L1 | sorghum root, chestnut wood | *Micrococcus luteus* | KX447587 | 1 | 0.00 | 0.00 |

**Table 1.** *Cont.*

| Strain | Origin | Taxonomic Identification | GenBank Accession Number | IAA [a] (Intensity Colour Scale 0–5) | Siderophore (HD/CD [b] cm) | Phosphate Solubilisation (DCP [c], HD/CD cm) |
|---|---|---|---|---|---|---|
| PVB1S1 | tomato shoot, pasture soil | *Bacillus safensis* | KX440184 | 0 | 0.00 | 1.54 |
| PVB1S2 | tomato shoot, pasture soil | Bacillaceae bacterium | KX442613 | 0 | 0.00 | 1.50 |
| PVB1S3 | tomato shoot, pasture soil | *Acinetobacter lwoffii* | KX440393 | 0 | 0.00 | 0.00 |
| PVB1S5 | tomato shoot, pasture soil | *Bacillus cereus* | KX440517 | 0 | 0.00 | 0.00 |
| PVB1L1 | tomato leaf, pasture soil | *Bacillus thuringiensis* | KX442563 | 0 | 0.00 | 0.00 |
| PVB1L2 | tomato leaf, pasture soil | *Bacillus muralis* | KX437755 | 0 | 5.04 | 0.00 |
| PVB1L3 | tomato leaf, pasture soil | *Bacillus cereus* | KX437756 | 0 | 0.00 | 1.37 |
| PVB1L5 | tomato leaf, pasture soil | *Bacillus cereus* | KX438058 | 0 | 0.00 | 0.00 |
| PVB1L6 | tomato leaf, pasture soil | *Bacillus megaterium* | KX438317 | 0 | 2.92 | 0.00 |
| PVB1L7 | tomato leaf, pasture soil | *Bacillus tequilensis* | KX438316 | 0 | 2.92 | 1.96 |
| PVB1L8 | tomato leaf, pasture soil | *Bacillus cereus* | KX438379 | 0 | 0.00 | 0.00 |
| PVB1R1 | tomato root, pasture soil | *Bacillus aerophilus* | KX440185 | 0 | 0.00 | 0.00 |
| PVB6R2 | tomato root, pasture soil | *Bacillus muralis* | KX440975 | 0 | 4.96 | 0.00 |
| PVB6R3 | tomato root, pasture soil | *Bacillus thuringiensis* | KX443415 | 0 | 0.00 | 1.42 |
| PVB6R4 | tomato root, pasture soil | Bacillaceae bacterium | KX443572 | 0 | 0.00 | 0.00 |
| PVB6R5 | tomato root, pasture soil | *Bacillus thuringiensis* | KX443561 | 0 | 0.00 | 0.00 |
| PVB6L1 | tomato leaf, pasture soil | *Acinetobacter johnsonii* | KX440619 | 2 | 0.00 | 0.00 |
| PVB6L2 | tomato leaf, pasture soil | *Microbacterium schleiferi* | KX440956 | 0 | 0.00 | 1.79 |
| PVB6L3 | tomato leaf, pasture soil | *Acinetobacter johnsonii* | KX440957 | 1 | 0.00 | 0.00 |
| PVB6L4 | tomato leaf, pasture soil | *Bacillus cereus* | KX440973 | 0 | 0.00 | 1.55 |
| PVB7R1 | tomato root, pasture soil | *Bacillus subtilis* | KX442589 | 0 | 1.58 | 2.00 |
| PVB7R2 | tomato root, pasture soil | *Bacillus cereus* | KX442609 | 0 | 0.00 | 0.00 |
| PO2R2 | tomato root, vegetable garden | *Paenibacillus* sp. | KX442611 | 0 | 0.00 | 0.00 |
| PO2L1 | tomato leaf, vegetable garden | *Bacillus niacini* | KX437751 | 0 | 0.00 | 0.00 |
| PO3S1 | tomato shoot, vegetable garden | *Bacillus thuringiensis* | KX442562 | 0 | 0.00 | 0.00 |
| PO7S1 | tomato shoot, vegetable garden | *Kochuria palustris* | KX442561 | 0 | 1.83 | 1.61 |

[a] Indole-3- Acetic Acid (IAA); [b] HD/CD refers to the halo diameter ratio; [c] Dicalcium phosphate (DCP). DCP solubilization was indicated by a clarification halo around the colony, while the capability to solubilize TCP was identified by bacterial growth on the medium.

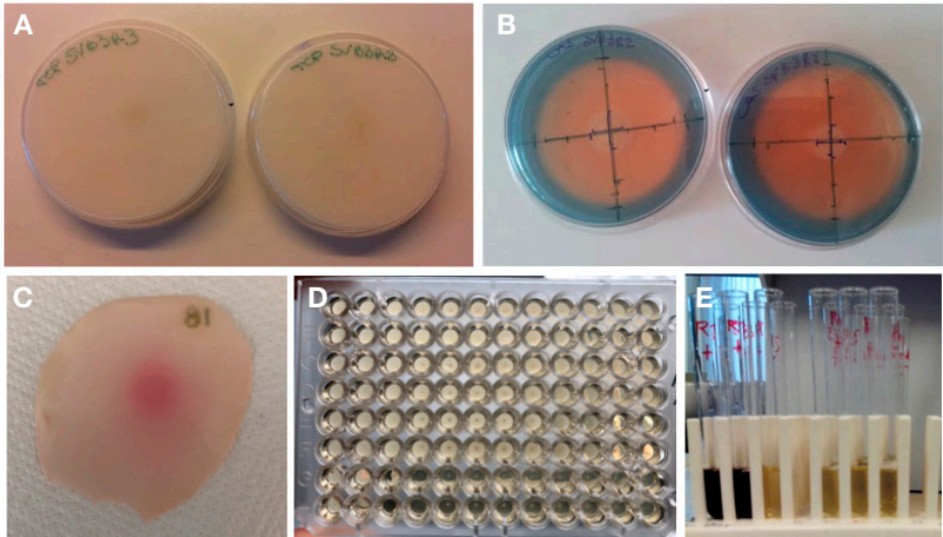

**Figure 3.** Screening of the physiological traits involved in plant growth promotion. (**A**) Phosphate solubilization on tricalcium phosphate (TCP) medium (bacterial growth indicates phosphate solubilizing activity); (**B**) siderophore synthesis on Chrome Azurol S (CAS) agar (orange halo); (**C**) IAA production (pink halo); (**D**) tolerance to salt stress evaluated as minimal inhibitory concentration (MIC) showing the tolerance of bacterial strains to specific molecules. The well was considered positive when the presence of a bacterial cell pellet was observed; (**E**) 1-aminocyclopropane-1-carboxylate (ACC) deaminase synthesis (brown/black color development).

Based on the taxonomic identification (possible plant and human pathogens have been excluded) and the results of the physiological activity assay (at least one positive character), two bacterial strains isolated from cucumber (*Herbaspirillum lusitanum* CVB2R5, *Stenotrophomonas rhizophila* CVB3S5) and six (*Pseudomonas migulae* SVB3R2, *P. migulae* SVB3R3 *P. migulae* SVB3R4, *Pseudomonas* sp. SVB3R5, *P. brassicacearum* SVB6R1, *Pseudomonas* sp. SL3R6) from sorghum were selected for further analysis.

The characterization of the antibiotic resistance profile revealed low levels of antibiotic resistance in the eight strains (Table 2). All bacterial endophytes were resistant to nitrofurans. Besides that, *H. lusitanum* CVB2R5 was resistant to phosphomycin, *P. brassicacearum* SVB6R1 to cefixime, and *S. rhizophila* CVB3S5 to both cefixime and cefoperazone, two third generation cephalosporins.

In order to assess whether the eight strains were true endophytes, we selected spontaneous rifampicin-resistant mutants from each strain and used them for the re-inoculation test. Both strains isolated from cucumber (*Herbaspirillum lusitanum* CVB2R5 and *Stenotrophomonas rhizophila* CVB3S5) were unable to colonize the leaves, the shoot or the root tissues of this plant species. On the other hand, all strains isolated from sorghum, except for *Pseudomonas* sp. SL3R6, were found in internal plant tissues; all five strains were found inside roots and leaves, except for *P. migulae* SVB3R4 which was unable to colonize leaves. In general, as reported in Table 3, the bacterial density (evaluated as CFU/g of root or shoot dry weight) was about $1 \times 10^3$.

**Table 2.** Sensitivity (S) or resistance (R) of eight selected bacterial endophytes to different antibiotics.

| | *H. lusitanum* CVB2R5 | *S. rhizophila* CVB3S5 | *P. migulae* SVB3R2 | *P. migulae* SVB3R3 | *P. migulae* SVB3R4 | *Pseudomonas* sp. SVB3R5 | *P. brassicacearum* SVB6R1 | *Pseudomonas* sp. SL3R6 |
|---|---|---|---|---|---|---|---|---|
| Ceftazidime 30 µg | S | S | S | S | S | S | S | S |
| Cotrimoxazole 25 µg | S | S | S | S | S | S | S | S |
| Gentamicin 10 µg | S | S | S | S | S | S | S | S |
| Ciprofloxacin 5 µg | S | S | S | S | S | S | S | S |
| Nalidixic acid 30 µg | S | S | S | S | S | S | S | S |
| Nitrofurans 100 µg | R | R | R | R | R | R | R | R |
| Cefoperazone30 µg | S | R | S | S | S | S | S | S |
| Phosphomycin 50 µg | R | S | S | S | S | S | S | S |
| Cefixime 10 µg | S | R | S | S | S | S | R | S |
| Norfloxacin 10 µg | S | S | S | S | S | S | S | S |

**Table 3.** Bacterial density inside sorghum plants following plant re-inoculation with the different strains.

| Bacterial Strain | Log CFU/g Plant Tissue | |
| --- | --- | --- |
| | **Root** | **Leaves** |
| *Pseudomonas migulae* SVB3R2 | 3.48 ± 1.01 | 3.76 ± 1.19 |
| *Pseudomonas migulae* SVB3R4 | 3.47 ± 1.15 | n.d. |
| *Pseudomonas brassicacearum* SVB6R1 | 3.18 ± 1.11 | 3.43 ± 1.09 |
| *Pseudomonas migulae* SVB3R3 | 3.37 ± 1.00 | 3.53 ± 1.02 |
| *Pseudomonas* sp. SVB3R5 | 3.63 ± 1.25 | 2.83 ± 1.07 |
| *Pseudomonas* sp. SL3R6 | n.d. | n.d. |

n.d.: not detected. CFU: colony forming units.

According to the ability of the isolated strains to colonize plant tissues after inoculation, we selected five strains (*P. migulae* SVB3R2, *P. migulae* SVB3R3, *P. migulae* SVB3R4, *Pseudomonas* sp. SVB3R5 and *P brassicacearum* SVB6R1) as true bacterial endophytes and used them for further experiments. All five strains increased plant and root biomass compared to uninoculated plants (Figure 4A,B). *P. migulae* SVB3R3 and *P. brassicacearum* SVB6R1 were the most effective, increasing the weight of whole plants by 54.5% and 53.9%, respectively and root weight by 62.5% and 65.9%, respectively. Shoot fresh weight was higher in plants treated with *P. migulae* SVB3R2 (+37.4%) and SVB3R3 (+43.3%) (Figure 4C), while all strains except SV6R1 enhanced the number of leaves (Figure 4D) compared to controls.

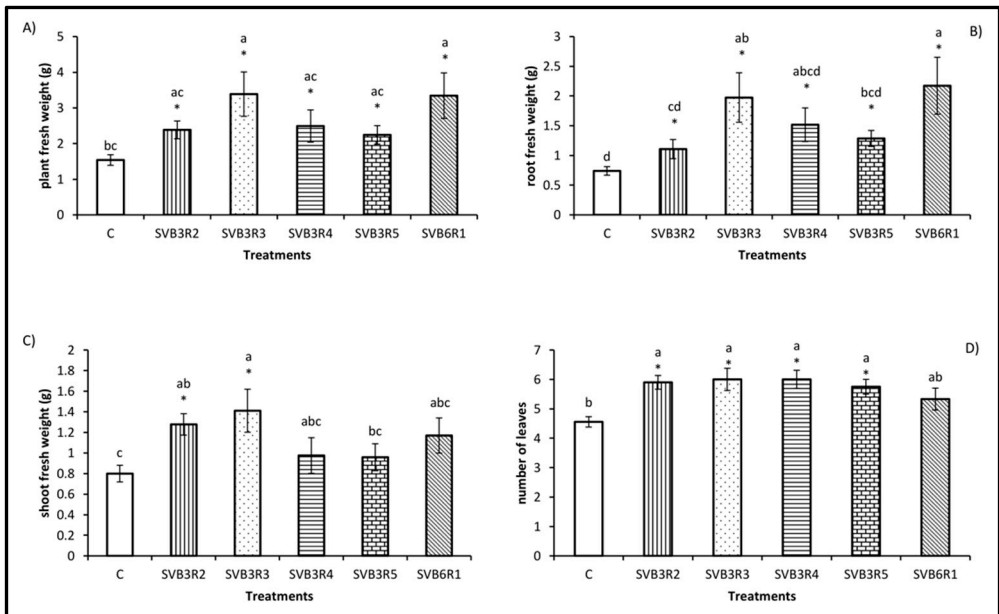

**Figure 4.** Effects of the selected bacterial strains on sorghum growth parameters (plant fresh weight, (**A**); root fresh weight, (**B**); shoot fresh weight, (**C**); and leaf number, (**D**)). The values presented in the figures are means ± standard errors (*n* = 10). Bars topped by the same letter do not differ significantly at $p < 0.05$ by one-way ANOVA followed by Fisher's post hoc test. Bars topped by * are significant at $p < 0.05$ by one-way ANOVA followed by Fisher's post hoc test compared to uninoculated control plants. Plant treatments: uninoculated plants (C), Plants inoculated by *P. migulae* SVB3R2 (SVB3R2), *P. migulae* SVB3R3 (SVB3R3), *P. migulae* SVB3R4 (SVB3R4), *Pseudomonas* sp. SVB3R5 (SVB3R5) and *P. brassicacearum* SVB6R1 (SVB6R1).

Three of the selected strains, *P. brassicacearum* SVB6R1, *P. migulae* SVB3R3 and *P. migulae* SVB3R4, were resistant to inhibition by 7.2% salt; the MIC values for *P. migulae* SVB3R2 and *Pseudomonas* sp. SVB3R5 were 5.4% and 6.2%, respectively. Therefore, the salt tolerance conferred by bacterial strains was largely above the salt concentration applied for plant treatment.

Considering all the physiological activities as well as the high tolerance to salt (Table 1) expressed by the five bacterial endophytic strains, we decided to assess their ability to increase plant tolerance to salt stress. In a preliminary experiment, performed by treating the plants with 0, 50, 100, 150 and 200 mM NaCl, the 150 mM concentration was found to be the sublethal concentration. All plant parameters, especially shoot weight, were reduced by feeding the plants with a nutrient solution containing 150 mM NaCl. Salinity stress reduced plant biomass by 89.5% (Figure 5A). *P. migulae* SVB3R4 was the most efficient strain in promoting the growth of the plant exposed to salinity: the plant biomass (+68.3%) as well as the root (+56.6%), shoot (+73.1%) fresh weight and leaf number (+28.5%) were significantly higher in plants inoculated with SVB3R4 than in uninoculated plants fed with 150 mM salt (Figure 5A–D). Moreover, strain *P. migulae* SVB3R3 also significantly increased the root weight (+52.1%), compared to controls exposed to salt (Figure 5B).

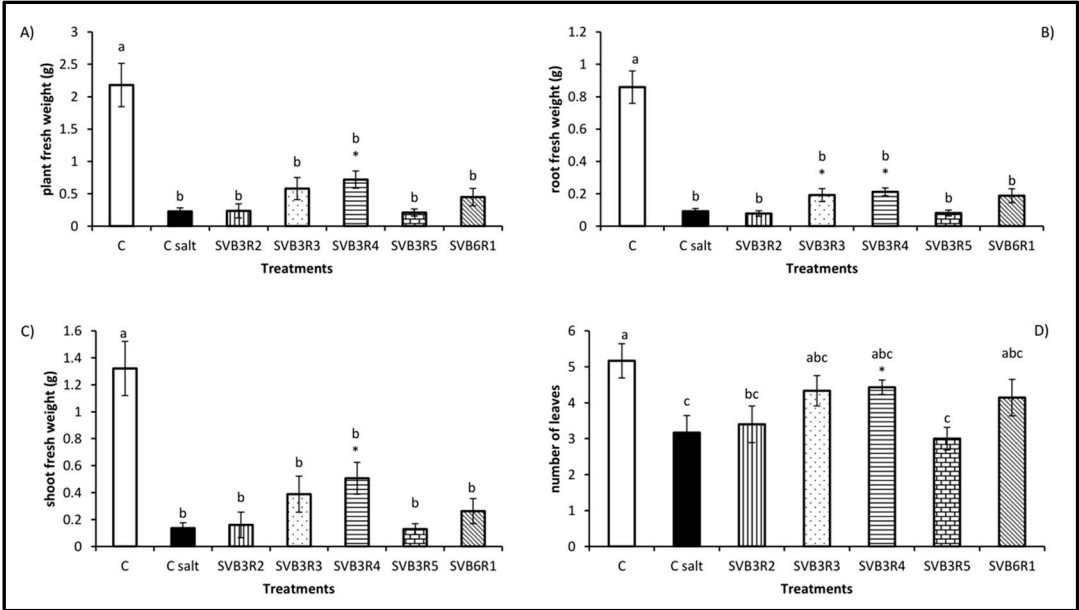

**Figure 5.** Effects of the selected bacterial strains on growth parameters (plant fresh weight, (**A**); root fresh weight, (**B**); shoot fresh weight, (**C**); and leaf number, (**D**)) of sorghum grown in the presence or in absence of 150 mM NaCl. The values presented in the figures are means ± standard errors (*n* = 7). Bars topped by the same letter do not differ significantly at *p* < 0.05 by one-way ANOVA followed by Fisher's post hoc test. Bars topped by * are significant at *p* < 0.05 by one-way ANOVA followed by Fisher's post hoc test compared to uninoculated control plants exposed to salt. Plant treatments: uninoculated plants (**C**), Plants inoculated by *P. migulae* SVB3R2 (SVB3R2), *P. migulae* SVB3R3 (SVB3R3), *P. migulae* SVB3R4 (SVB3R4), *Pseudomonas* sp. SVB3R5 (SVB3R5) and *P. brassicacearum* SVB6R1 (SVB6R1).

Symptom expression in uninoculated plants started on the fourth day of treatment and 42.9% of the plants died after 20 days of growth. In the same way, all plants inoculated with the bacterial endophytes, except those inoculated with *P. brassicacearum* SVB6R1, began to show minor symptoms on the sixth day. At the time of harvest, 80% of the plants inoculated with strain SVB3R2 were dead. In contrast, none of the plants inoculated with strains SVB3R3 and SVB3R4 died by the 20th day. However, 83% of the plants inoculated with SVB3R3 and 71.4% of those bacterized with SVB3R4 showed strong symptoms by the end of the experiment. Similarly, only 20% of the plants inoculated with SVB3R5 survived at the end of the experiment besides showing strong symptoms. Only 28.5% of the plants inoculated with *P. brassicacearum* SVB6R1 died by the last day of the experiment, while 28.5% and 14.3% of the plants showed weak and strong symptoms, respectively, starting from the 18th day. Moreover, 28.5% of the plants remained symptomless. Weak symptoms were observed in 42.8% of the

plants inoculated with strain SVB6R1, starting from the 12th day, thus indicating a significant delay in symptom expression compared to the uninoculated control (Figure 6).

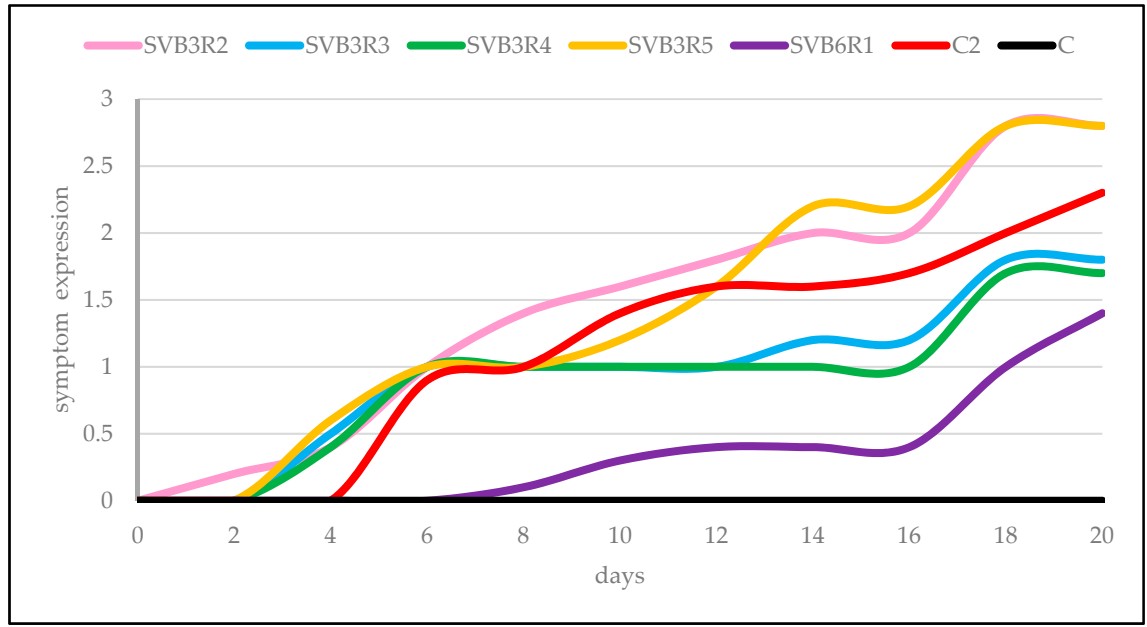

**Figure 6.** Symptom expression (mean values, *n* = 7) in inoculated and uninoculated plants exposed to 150 mM salt: 0 = absence of evident symptoms, 1 = weak symptoms (yellowing of the leaf tips), 2 = strong symptoms (leaves dried or completely yellow), and 3 = plant death. Bacterial treatments: *P. migulae* SVB3R2 (SVB3R2), *P. migulae* SVB3R3 (SVB3R3), *P. migulae* SVB3R4 (SVB3R4), *Pseudomonas* sp. SVB3R5 (SVB3R5) and *P. brassicacearum* SVB6R1 (SVB6R1).

The clear effects of the strains *P. migulae* SVB3R3 and *P. brassicacearum* SVB6R1 on the growth of plants exposed to salinity stress are shown in Figure 7A,B, respectively. Since ACC deaminase activity is typically involved in increasing plant stress tolerance, we then decided to assess the occurrence of this physiological trait in the five bacterial endophyte strains. Only strain *P. brassicacearum* SVB6R1 contained a moderate level of this enzymatic activity (11.9 µmol alpha ketobutyrate/mg protein/hour).

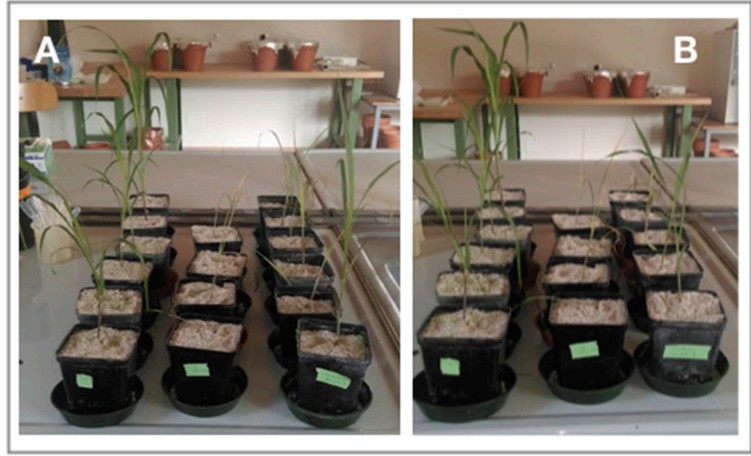

**Figure 7.** Effects of the strains *P. migulae* SVB3R3 (**A**) and *P. brassicacearum* SVB6R1 (**B**) on the growth of plants exposed to salt stress. From left to right: control plants not exposed to NaCl, plants fed with 150 mM NaCl and endophyte-inoculated plants fed with 150 mM NaCl.

## 4. Discussion

In this work, three non-perturbed soils subjected to different management regimes were considered as a possible source for bacterial endophytic strains with potential in plant growth promotion. Moreover, three economically relevant plant species (tomato, cucumber and sorghum) were used as traps for endophytes. Characterization of plant beneficial traits showed that each considered plant species hosted a plethora of endophytes with multiple plant growth promotion abilities, some of which could significantly enhance seed germination, as well as plant growth and development.

This kind of physiological characterization allowed us to select some strains that can be considered potentially efficient biofertilizers and to highlight the relationships between bacteria with specific physiological activities and a particular plant species. In fact, our results showed that 56% of the endophytes isolated from cucumber displayed indole 3-acetic acid (IAA) production, 55% of the strains from tomato showed phosphate solubilization activity, and 42% of the strains from sorghum showed siderophore production. Moreover, the occurrence of bacterial endophytes able to synthesize IAA, solubilize phosphate and produce siderophores was highest in pasture soil (55.5%, 76.2% and 57.1%, respectively). In particular, our results demonstrated that pasture soil, the least perturbed soil among those considered, is probably the one with the highest biodiversity and richness in bacterial species. For this reason, the highest number of endophytic strains with plant beneficial physiological traits were isolated from this soil. This picture is obviously restricted to culturable endophytic bacteria, which are likely a very small fraction of the inner microbiota of the plant, that in turn represent an even smaller portion of the total rhizosphere microbiota. However, it is well established in the literature that plant genotype drives not only the development of plant phenotypes, but also the establishment of a specific plant microbiota [31]. In addition, the type of soil used to cultivate a plant species can also shape its endophytic community. Consistently, different endophytic populations can be recorded in the same plant species growing in different soils [22]. About 27% of the isolates showed the ability to synthesize IAA. Production of IAA in bacteria occurs though six different metabolic pathways, with five of them based on tryptophan as a precursor [32,33]. Bacterial IAA affects the plant endogenous auxin pool; as a consequence, the amount of endogenous auxin is often the determinant of the bacterial IAA effect on plant growth. High levels of total auxin enhance ethylene synthesis in plants, leading to leaf and fruit abscission and inhibition of stem growth. In this way, IAA synthesized by bacteria is directly related to plant growth and development [34].

The low bioavailability of P and Fe in most soils is one of the main factors limiting plant growth [34]. Therefore, bacteria able to solubilize and mineralize P (phosphate-solubilizing bacteria, PSB) and synthesize siderophores have great potential as biofertilizers. In this work, 30% of the bacterial endophytes isolated were able to solubilize phosphate and 32% were able to produce siderophores. Solubilization of inorganic P (Pi) by PSB occurs via the production of low molecular weight organic acids, leading to acidification of soil and then to increased solubilization of inorganic phosphate [35]. The evidence showing that PSB can directly affect plant Pi acquisition inside the root is still missing [36]. However, a linear relationship between the phosphate solubilization activity of PSB and increased phosphorus levels in tissues of plants inoculated with PSB was reported by Chabot et al. [37]. Similarly, iron bioavailability in aerobic soil is very low, ranging from about $1 \times 10^{-7}$ at pH 3.5 to $1 \times 10^{-23}$ M at pH 8.5 [38]. Both microbes and plants have a high iron requirement (about $1 \times 10^{-5}$–$1 \times 10^{-7}$ and $1 \times 10^{-4}$–$1 \times 10^{-9}$ M, respectively) and this need is even more accentuated in the rhizosphere, where the competition for this element among plant roots, bacteria and fungi is very intense [39]. Bacteria react to iron deficiency by siderophore synthesis [40] and are involved in both direct (stimulation of plant growth by improving iron nutrition) and indirect effects (stimulation of plant growth by phytopathogen inhibition) on plant growth. The positive effects of siderophores synthesized by bacteria on plant growth have been demonstrated by Vansuyt et al. [41]. In this work, *Arabidopsis thaliana* plants were supplied with ferri-siderophores produced by *Pseudomonas fluorescens* C7 as a sole source of iron, with the result that the bacterial Fe–pyoverdine complex was efficiently taken up by the plants and an increased iron content inside plant tissues associated with the improved plant growth was observed [41]. On the other

hand, siderophores produced by PGPB limit iron acquisition by phytopathogenic fungi, which are unable to produce iron chelators with same the high affinity for iron as the siderophores produced by PGPB. Plant pathogen suppression by bacterial siderophores has been reported in the biocontrol of *Macrophomina phaseolina*, *Rhizoctonia solani*, *Pythium* spp., and *Fusarium* spp. [40]. Additionally, bacterial siderophores may trigger induced systemic resistance (ISR) in plants, thereby enhancing plant resistance to phytopathogen infection [42] as observed in rice infected by *Magnaporthe oryzae* and treated with a pseudobactin siderophore synthesized by *Pseudomonas fluorescens* WCS374r. Plants inoculated with a mutant of this bacterial strain unable to produce pseudobactin were no longer resistant to growth inhibition by this pathogen [43].

Based on the taxonomic identification and the physiological traits involved in plant growth stimulation, eight endophytic bacterial strains (two isolated from cucumber and six from sorghum) were selected for further analysis.

In view of a future possible application of these strains as biofertilizers in field conditions, it was considered essential to assess their antibiotic resistance profile. In fact, in the environment and especially in soil, antibiotic resistance genes may be acquired through horizontal gene transfer from innocuous bacterial species to species considered as potentially pathogenic [44]. Due to the extensive use of antibiotics in animal farming in recent decades, soil represents a sink of antibiotic resistant human pathogens [45–48]. Antibiotic resistance in opportunistic pathogens is a serious concern for human and animal health and is strictly associated with increased hospitalization and mortality rates [49,50]. Moreover, antibiotic resistant pathogenic bacteria, as well as their antibiotic resistance genes, can be transferred from organic fertilizers and manure to vegetables and fruits via irrigation [51,52]. It has been recently reported [44] that antibiotics can be taken up by plant roots, and then translocated to the fruit and leaves, where a sub-lethal amount of antibiotics (such as tetracycline, streptomycin, and ciprofloxacin) has been detected. In addition, plant internal colonization by opportunistic human pathogens occurs in different crops [53,54]. In this regard, none of the eight selected strains belonged to species reported to be a human opportunist pathogen and they all showed a low level of resistance to the antibiotics considered in this work.

All eight strains were used in order to assess their effective plant internal colonization ability. Only five strains were detected after re-inoculation inside sorghum roots and/or leaves, with a density of around $1 \times 10^3$ CFU/g plant tissue. This is in agreement with the literature reporting a density of endophytic bacteria ranging from $1 \times 10^2$ to $1 \times 10^9$ per gram of plant tissue depending on the plant species, the plant organ considered, the phenological stage of the plant, the plant cultivar and the interaction with other organisms, as well as other environmental factors [55]. These five strains belonged to *Pseudomonas migulae*, *P. brassicacearum* and *Pseudomonas* sp. Interestingly, other strains belonging to these bacterial species have been previously reported as endophytic [22,56,57].

All selected bacterial strains were able to improve plant development for at least one of the growth parameters considered. Therefore, in general, all strains behave as plant growth promoters on sorghum. In more detail, the SVB3R3 strain (able to synthesize siderophores and solubilize phosphate, but unable to produce IAA) significantly stimulated all plant parameters, while the SVB6R1 strain (positive for all three physiological traits tested) increased only the biomass of the plant and the weight of the roots. This suggests that in vitro tests of bacterial physiological activities are not always able to define the activity of the bacterial strain in vivo. Moreover, it must be considered that other unknown mechanisms may be involved in promoting plant growth.

Sorghum is one of the best adapted cereals to water limited soils and is classified as one of the most drought tolerant crops cultivated [58]. However, it was shown that huge variations in sorghum salt tolerance exist among the different cultivated genotypes [59]. Therefore, possible support from bacteria able to enhance stress tolerance in plants could be desirable. The effects of drought and salinity stress on plants partially overlap and impair several plant physiological processes, such as seed germination, seedling growth and vigor, and flowering and fruit production [60]. Numerous reports have indicated the involvement of soil bacteria in increasing plant tolerance to the inhibitory effects

of salt. However, only a small fraction of those reports deal with bacterial endophytes. For example, Rho et al. [61] performed a meta-analysis on plant stress mitigation by endophytes based on the results presented in 209 papers. This statistical analysis showed that endophytes inducing plant stress alleviation belonged to bacteria (53%), filamentous fungi (41%) and yeast (6%). The results showed the positive impacts of endophytes on plants exposed to drought, nitrogen deficiency, and salinity stress conditions. In agreement with previous observations, the growth of the uninoculated plants treated with a salt solution was dramatically reduced in our study. However, strain SVB3R4 increased all measured plant growth parameters. Consistent with this result, Eida et al. [62], showed that 11 endophytic bacterial isolates from Jizan (Saudi Arabia) exhibited plant growth promotion abilities on *Arabidopsis thaliana* exposed to 100 mM NaCl. Similarly, bacterial endophytes *P. fluorescens* YsS6 and *P. migulae* 8R6 were shown to enhance plant growth in non-stressed condition and subsequently protect tomato plants against salinity (165 and 185 mM) by facilitating their growth under such a stressful condition [63]. Interestingly, in plants inoculated with rhizospheric bacteria (model strain *P. putida* UW4), excess salt was accumulated in the vacuole, a confined area far away from cytosolic metabolism [63]. On the contrary, plant growth-promoting bacterial endophytes were found to limit the concentration of sodium in plant shoots through a still unknown mechanism. The evaluation of the expression of salt stress related symptoms in sorghum plants indicated that SVB3R3 and SVB3R4 strains reduced the severity of symptoms across a narrow time range (7–14 days) and then returned to the level expressed by control plants. On the other hand, strain SVB6R1 delayed and reduced the expression of symptoms throughout the experiment, also increasing the survival of the plants.

Since ACC deaminase activity is usually involved in increasing plant stress tolerance, we decided to assess the occurrence of this physiological trait in the five bacterial endophyte strains. Only *P. brassicacearum* SVB6R1 showed a moderate level of this enzymatic activity, suggesting that, in this strain, ACC deaminase is involved in delaying and limiting the level of stress symptom expression in the host plant. However, a clear demonstration of the role of this enzyme requires the construction and testing of a mutant lacking this activity. Moreover, strains SVB3R3 and SVB3R4 were both able to reduce stress symptoms while being unable to produce either ACC deaminase or IAA; this indicates that other, still unknown mechanisms could be involved in increasing plant tolerance to salt stress.

The results reported here suggest that ACC deaminase is one, but not the only, bacterial trait involved in conferring plant tolerance to salt stress. However, many questions remain to be addressed before the possible application of these strains in sustainable agriculture. Thus, there is a gap in the molecular information regarding the communication between the endophytic strains and the host plants, the establishment of this association, and the modality of transmission of the bacterial endophyte.

## 5. Conclusions

In this work, we isolated about 60 bacterial endophytes from three economically relevant plant species (cucumber, tomato and sorghum) grown in three different soils with different managements (high mountain pasture soil, a kitchen garden soil and a chestnut wood soil). All of the bacterial strains were identified and characterized for some of the physiological activities potentially involved in plant growth promotion. We then selected eight strains that were tested for their capability to effectively colonize internal plant tissues and for their antibiotic resistance profile. Five of these bacterial endophytes were used as inoculants on sorghum plants and were demonstrated to behave as plant growth promoters. Since an efficient PGPB should also improve the development and health of plants facing environmental stresses, we assessed the impact of these five endophytic strains on sorghum plants exposed to salt stress. One of the strains increased plant growth parameters compared to uninoculated controls and was particularly efficient in delaying salt-related symptom expression in plants. Recently, the interaction between plants and endophytes has received considerable attention. The physiological activities of these bacteria are expressed directly inside the plants, making them

useful both as biofertilizers and biopesticides. Moreover, the broad host range showed by many bacterial endophytes makes them a powerful tool in the development of sustainable agriculture.

**Author Contributions:** E.G., E.B. and N.F. conceived and planned the experiments, interpreted the results and worked on the manuscript. G.N. and N.F. carried out the experiments. E.B. and P.C. worked on bacterial molecular identification. B.R.G. and M.d.C.O.-M. performed the ACC deaminase test and reviewed the manuscript. G.B. and G.L. aided in the interpretation of the results and in finding funding. N.M. performed the statistical analyses. E.G. wrote the manuscript with input from all authors. All authors revised the manuscript. All authors have read and agreed to the published version of the manuscript.

**Funding:** This research received no external funding.

**Acknowledgments:** The authors are grateful to Donata Vigani for the technical help.

**Conflicts of Interest:** The authors declare no conflict of interest.

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
