# Peer review of "Screening of Bacterial Endophytes Able to Promote Plant Growth and Increase Salinity Tolerance"

_applsci, doi:10.3390/app10175767_

Round 1
Reviewer 1 Report
This manuscript is potentially interesting but has a lot of weaknesses. In general, its readability is very poor, the research design is not very clear, the results are poorly presented and confusing. Moreover, it is not very clear why some sets of data are included in the manuscript (for example, antibiotic resistance data).
Major comments
1) The manuscript is about the isolation of bacterial endophytes, subsequent characterization, and selection of strains able to promote plant growth and increase salinity tolerance. Selection criteria are of paramount importance for the research design. Unfortunately, nothing is said about the criteria, beside a vague statement that bacterial strains were selected 'based on the physiological trait analysis'. The authors must explain in detail how they chose the bacterial strains for further analysis.
2) Results are not clearly presented. Legends and figures do not contain all necessary information. For example, figures 2 and 3 do not mention replica numbers. In all figures and tables, abbreviations used are not explained in the legend and, in some cases, it is not possible to find the significance of the abbreviation in the text.
3) The presentation of statistical results in figures 2 and 3 is totally confusing. I do not understand the reason to use letters and * to represent the statistical significance. In the case of figure 3, this leads to contrasting results. According to *, there is a significant difference compared to control plants exposed to salt; on the contrary, according to letters, there is no significant difference. Which is the right one?
4) So, in the end, it is not clear whether bacterial strains are able to promote plant growth in the presence of salt stress.
5) The section Results and Discussion should be vastly reduced to improve the readability of the manuscript. It has no sense to 'review' the scientific literature on IAA, siderophore, and P/Fe if data are not discussed. My suggestion is to split this section into the Results section and the Discussion section.
Author Response
Reviewer 1
This manuscript is potentially interesting but has a lot of weaknesses. In general, its readability is very poor, the research design is not very clear, the results are poorly presented and confusing. Moreover, it is not very clear why some sets of data are included in the manuscript (for example, antibiotic resistance data).
Major comments
- Q: The manuscript is about the isolation of bacterial endophytes, subsequent characterization, and selection of strains able to promote plant growth and increase salinity tolerance. Selection criteria are of paramount importance for the research design. Unfortunately, nothing is said about the criteria, beside a vague statement that bacterial strains were selected 'based on the physiological trait analysis'. The authors must explain in detail how they chose the bacterial strains for further analysis.
A: The bacterial endophytic strains have been selected for further analysis according to their taxonomic identification (all the strains belonging to species that can be considered as phytopathogenic i.e. Agrobacterium, or human opportunistic pathogens have been eliminated), and to the expression of at least one plant beneficial trait. A sentence indicating this criterion has been added to the text.
- Q: Results are not clearly presented. Legends and figures do not contain all necessary information. For example, figures 2 and 3 do not mention replica numbers. In all figures and tables, abbreviations used are not explained in the legend and, in some cases, it is not possible to find the significance of the abbreviation in the text.
A: The replica numbers for Figures 2 and 3 (Figures 4 and 5 in the new version of the paper) has been added to the legends. We also added details regarding the abbreviations used.
3) Q: The presentation of statistical results in figures 2 and 3 is totally confusing. I do not understand the reason to use letters and * to represent the statistical significance. In the case of figure 3, this leads to contrasting results. According to *, there is a significant difference compared to control plants exposed to salt; on the contrary, according to letters, there is no significant difference. Which is the right one? So, in the end, it is not clear whether bacterial strains are able to promote plant growth in the presence of salt stress.
A: The statistical analysis has been performed among all the treatments and the significant values have been indicated with different letters. Moreover, in order to optimize the performance of the statistical analysis, we performed a comparison between each single bacterial treatment against the uninoculated control exposed to salt stress (this kind of comparison allowed us to increase the degree of statistical freedom); in this case, the significant values were indicated by a *. The strain SVB3R4 increased plant growth parameters compared to uninoculated control plants exposed to salinity; however, looking at the symptoms of salt stress, strain SVB6R1 was the most efficient in delaying and reducing the expression of these symptoms.
5) Q: The section Results and Discussion should be vastly reduced to improve the readability of the manuscript. It has no sense to 'review' the scientific literature on IAA, siderophore, and P/Fe if data are not discussed. My suggestion is to split this section into the Results section and the Discussion section.
A: The Authors accept the opinion of the reviewers. The Results and Discussion are now separated. All of the Discussion has been extensively reviewed and rewritten. We hope that this new version of the manuscript will meet your expectations.
Reviewer 2 Report
The paper is devoted to the screening of plant-beneficial physiological abilities of bacterial endophytes isolated from the tissues of three economically important plant species grown in different regions of Italy. The topic of the paper is really economically and ecologically important. The authors conducted a great volume of experimental and laboratory work and obtained interesting and promising results. But both presentation and analysis of these results need to be substantially improved.
Introduction
Lines 73-77: The goal (like screening the physiological traits of bacterial endophytes which can be beneficial for plant development) and the objectives of the study should be formulated more precisely. The authors should also listed the traits (characteristics) of bacterial endophytes analyzed in the course of the study (e.g., siderophore production, phosphate solubilization, synthesis of auxin and ACC deaminase, resistance to antibiotics, capacity to recolonize plant tissues, effect on plants exposed to salinity stress, etc.)
Material and Methods
M&M should start with the subsections devoted to site description (2.1; location, climate, main crops, etc.) and separately (2.2) - sampling and experimental design.
What does it mean – "three soils were collected"? Namely soil samples? For which purposes? It should be described how plant seeds were collected.
All abbreviations used must be explained at their first mention.
Results and Discussion
The authors did not divide this section into subsections devoted to different traits and abilities of bacterial endophytes, and it seriously complicated the reading and understanding of the data presented. In fact, the analysis of the results obtained is almost absent, and the discussion is presented practically only as a review of the previously published findings.
If the authors isolated bacterial endophytes from different plant species grown at different regions (soils), why they did not analyze the influence of host plants and the soil where they were grown as well as plant tissues (shoots, roots, leaves) on the endophyte composition and abilities? Were the endophytic traits studied species-specific, strain-specific or none-specific? Is it possible to compare the level of traits measured in the studied bacterial endophytes with the level of traits of the same kind revealed in previous investigations?
The paper must also contain the conclusion section summarizing most important findings and including the application perspectives.
The quality of the tables needs to be improved. Table 2 is colorful and does not contain any explanation of different colors. It is also unclear why the strains showed no activities were included in the table. Table 4 is presented in improper way. It should look like this:
Table 4. Density of bacterial endophytes (log CFU/g plant tissue ± SR) in plant tissues of sorghum after reinoculation(?)
|
Bacterial strain |
Plant tissue |
|
|
roots |
leafs |
|
|
Pseudomonas migulae SVB3R2 |
3.48 ± 1.01 |
3.76 ± 1.19 |
The language of the manuscript needs to be thoroughly checked and corrected.
All other numerous corrections and suggestions are inserted into the PDF version of the manuscript, which is attached.

Author Response
Reviewer 2
The paper is devoted to the screening of plant-beneficial physiological abilities of bacterial endophytes isolated from the tissues of three economically important plant species grown in different regions of Italy. The topic of the paper is really economically and ecologically important. The authors conducted a great volume of experimental and laboratory work and obtained interesting and promising results. But both presentation and analysis of these results need to be substantially improved.
Introduction
Q: Lines 73-77: The goal (like screening the physiological traits of bacterial endophytes which can be beneficial for plant development) and the objectives of the study should be formulated more precisely. The authors should also listed the traits (characteristics) of bacterial endophytes analyzed in the course of the study (e.g., siderophore production, phosphate solubilization, synthesis of auxin and ACC deaminase, resistance to antibiotics, capacity to recolonize plant tissues, effect on plants exposed to salinity stress, etc.)
A: We agree with the reviewer and we have modified the text accordingly.
Material and Methods
Q: M&M should start with the subsections devoted to site description (2.1; location, climate, main crops, etc.) and separately (2.2) - sampling and experimental design.
A: We agree with the reviewer and we have separated the two paragraphs into 2.1 Sampling site and 2.2 soil sampling and experimental design (we added more information about the soils and climate). Moreover, we provided a Figure indicating the geo-localization of the three sites (Figure 1) and one showing climatic data (Figure 2).
Q: What does it mean – "three soils were collected"? Namely soil samples? For which purposes? It should be described how plant seeds were collected.
A: We modified the text according to this request and have explained the soil sampling procedure in some detail.
The soil sample
Q: All abbreviations used must be explained at their first mention.
A: This has been done as suggested.
Results and Discussion
Q: The authors did not divide this section into subsections devoted to different traits and abilities of bacterial endophytes, and it seriously complicated the reading and understanding of the data presented. In fact, the analysis of the results obtained is almost absent, and the discussion is presented practically only as a review of the previously published findings.
A: As suggested by the reviewer, the Results and Discussion are now separated. The Discussion section has been extensively rewritten.
Q: If the authors isolated bacterial endophytes from different plant species grown at different regions (soils), why they did not analyze the influence of host plants and the soil where they were grown as well as plant tissues (shoots, roots, leaves) on the endophyte composition and abilities? Were the endophytic traits studied species-specific, strain-specific or none-specific? Is it possible to compare the level of traits measured in the studied bacterial endophytes with the level of traits of the same kind revealed in previous investigations?
A: We thank the reviewer for this suggestion which has been taken into account in the discussion section of the new version of the manuscript. We hope the modifications we made will meet your expectations.
Q: The paper must also contain the conclusion section summarizing most important findings and including the application perspectives.
A: A conclusion section summarizing the most important points has been added to the end of the manuscript.
Q: The quality of the tables needs to be improved. Table 2 is colorful and does not contain any explanation of different colors. It is also unclear why the strains showed no activities were included in the table.
A: We eliminated the colors in the Table 2, that indicated the different origins of the strains. Moreover, we incorporated much of the data previously presented in Table 2 inside Table 1. We prefer to maintain strains without any positive activities because they are part of our bacterial population and we think it’s honest and correct to showing all of the results that were obtained.
Q: Table 4 is presented in improper way. It should look like this:
Table 4. Density of bacterial endophytes (log CFU/g plant tissue ± SR) in plant tissues of sorghum after reinoculation(?)
|
Bacterial strain |
Plant tissue |
|
|
Roots |
leafs |
|
|
Pseudomonas migulae SVB3R2 |
3.48 ± 1.01 |
3.76 ± 1.19 |
|
|
|
|
A: Table 4 is now Table 3, and it has been modified it as suggested.
Q: The language of the manuscript needs to be thoroughly checked and corrected.
A: The manuscript has been thoroughly reviewed by a native English speaker
All other numerous corrections and suggestions are inserted into the PDF version of the manuscript, which is attached.
Attached pdf
A: All of the minor corrections and suggestions made by the reviewer have been addressed in the revised version.
Line 19-20 abstract The authors prefer to maintain this sentence as is.
Legend of Figure 4 (previously Figure 2): The statistical analysis has been performed among all the treatments and the significant values were indicated with different letters. Moreover, in order to optimize the performance of the statistical analysis we performed a comparison between each single bacterial treatment against the uninoculated control exposed to salt stress (this kind of comparison allowed us to increase the degree of statistical freedom); in this case the statistically significant values were indicated by an *.
Data regarding the symptom expression of plants following salt stress presented in Figure 4 (Figure 6 in the new version of the manuscript) are the mean values of the symptomatic class of each single plant.
Reviewer 3 Report
The manuscript by Gamalero et al., on “Looking for bacterial endophytes able to promote 3 plant growth and increase salinity tolerance” deals with isolating endophytic bacteria of relevant crop plants and use it for plant growth promotion on one hand and to enhance the salinity tolerance of the agricultural plants using them on the other hand. Although the approach mentioned in this manuscript is of great relevance for betterment of agricultural practices and to combat salt stress of economically relevant crops, I find the planning for the experiments described in this manuscript not very rational and appropriate.
Nothing is mentioned in detail about the three soils with different managements. Why were they chosen? Do any of these different soil management regimes also experience high salinity? What crop plants are growing in these soil(s)? That would have been probably more relevant to isolate endophytes from crop plants still growing poorly in soils with high salt.
A 16S molecular phylogram of the relevant five Pseudomonas strains used for salinity tolerance testing of the host plant, Sorghum would have provided a more valuable and interesting information.
Part of the plant growth promoting traits of the isolated endophytes are very subjective (less quantitative) and poorly presented in this manuscript. The Presentation of data in several of the Tables are not satisfactory.
line 97: NaClO ??
line 221: Table 2: must be written in the beginning
line 226: screening of traits: more quantitative bar diagrams expected.
Line 391-392: correct the sentence
Author Response
Reviewer 3
The manuscript by Gamalero et al., on “Looking for bacterial endophytes able to promote 3 plant growth and increase salinity tolerance” deals with isolating endophytic bacteria of relevant crop plants and use it for plant growth promotion on one hand and to enhance the salinity tolerance of the agricultural plants using them on the other hand. Although the approach mentioned in this manuscript is of great relevance for betterment of agricultural practices and to combat salt stress of economically relevant crops, I find the planning for the experiments described in this manuscript not very rational and appropriate.
Q: Nothing is mentioned in detail about the three soils with different managements. Why were they chosen? Do any of these different soil management regimes also experience high salinity? What crop plants are growing in these soil(s)? That would have been probably more relevant to isolate endophytes from crop plants still growing poorly in soils with high salt.
A: In the Materials and Methods section we separated the two paragraphs: 2.1 Sampling site and 2.2 Soil sampling and experimental design. Moreover, we provided new figures with the geo-localization of the three sites (Figure 1) and the climatic data (Figure 2). We sampled these soils because (as stated in the revised text) they are “natural, non-perturbed soils” subjected to different management. Our first goal was to isolate bacterial endophytes from non-stressed soil in order to assess their ability to stimulate plant growth. The idea of testing these strains for their ability to increase tolerance to environmental stress came only later, also considering that sorghum is a drought resistant cereal but the range of this tolerance varies greatly among cultivars. We agree that in retrospect, it would have been theoretically more appropriate to start from saline soils, however, the high tolerance to salt concentration shown by the isolated bacteria, evaluated through the measurement of the MIC, indicate that the characteristics of the soil of origin do not totally define the physiological abilities of the resident bacteria.
Q: A 16S molecular phylogram of the relevant five Pseudomonas strains used for salinity tolerance testing of the host plant, Sorghum would have provided a more valuable and interesting information.
A: We agree with the reviewer, but we think that the data obtained by this kind of analysis can be part of another work. In this context, we only needed to get information about the taxonomic identification in order to eliminate possible human/plant pathogens from the screening.
Q: Part of the plant growth promoting traits of the isolated endophytes are very subjective (less quantitative) and poorly presented in this manuscript. The Presentation of data in several of the Tables are not satisfactory.
A: The Tables 1 and 2 in this new version of the paper, have been incorporated in order to improve the readability. Table 4 is now Table 3 and we modified it as requested by the other reviewers.
Q: line 97: NaClO ??
A: This is the chemical formula for sodium hypochloride.
Q: line 221: Table 2: must be written in the beginning
A: We have now incorporated Table 2 inside Table 1 and we changed the corresponding title.
Q: line 226: screening of traits: more quantitative bar diagrams expected.
A: In this study we used qualitative assessment in order to check for the presence or absence of several physiological traits. In this way, we obtained a semi-quantitive evaluation of the bacterial activities, that we think is not correct to represent as diagrams.
Q: Line 391-392: correct the sentence
A: This has been done as suggested.
Round 2
Reviewer 2 Report
The paper looks better, but in my strong opinion, in its present form it is not yet ready for the publication in the journal.
The second review round arose new questions and showed that some questions from the first round of the review had remained unanswered.
The authors insisted on the idea that they had selected three types of soil in order "to isolate new bacterial endophytes" (lines 19, 72, and 81). However, by definition, endophytes are inhabitants of the plant endosphere and cannot be isolated from a soil. If the authors mean that the soil is a possible source of bacterial endophytes and because of that, they had collected soil of different management type for growing plants in a greenhouse - they should state it clearly.
In Material and Methods, it should be described how the seeds for plant growing were collected.
Lines 148-151. The description of medium content is not clear because the concentration of main medium components - DCP and TCP, are not specified.
Lines 166-169. Once again, it should be specified, why namely this set of antibiotics was used. If it is a standard procedure to test the antibiotic resistance profile of a bacterial strain, with the standard set of antibiotics in standard concentrations - the relevant reference should be given.
Lines 254 and 416. "Based on the taxonomic identification" - the role of taxonomic identification in the selection of bacterial endophytes for the analyses is unclear. Does it mean that the strains belonging to specific species/genera were selected? It does not follow from what is written in M&M and Results.
Figure 6 – the question still remains: if 0 = absence of evident symptoms, what does -0.2 mean?
Once again, both Results and Discussion are not divided into subsections (according to the physiological traits, for example), and this complicates reading and understanding the text.
Once again, Discussion does not contain a sufficient analysis of the data presented. Except for the review of the previously published findings, it mostly just repeats the results obtained (as well as Conclusion). Thus, it is unclear for what purposes the authors used different plant species, grew them in the soil under different types of management, and isolated bacterial endophytes from different plant tissues. Just in order to increase the diversity of endophytes? It is OK. But why not to consider and analyze the influence of host plants (cucumber, sorghum, and tomato) and the soil where they were grown, as well as plant tissues (shoots, roots, leaves) on the endophyte composition and physiological abilities? For example, why namely the plants grown in the pasture soil gave the highest number of endophytes? Maybe because of the preferential soil physico-chemical properties?
In my opinion, for a research paper it is not enough just to answer the question "what", it should also explain "how" and "why". The manuscript in its current state does not contain such sufficient explanations.
Still, the language of the manuscript needs to be checked and corrected.
All other corrections and suggestions are inserted into the attached PDF version of the manuscript.

Author Response
Response to Reviewer 2
The Authors would like to thank you for your work in order to improove aour paper.
The paper looks better, but in my strong opinion, in its present form it is not yet ready for the publication in the journal.
The second review round arose new questions and showed that some questions from the first round of the review had remained unanswered.
The authors insisted on the idea that they had selected three types of soil in order "to isolate new bacterial endophytes" (lines 19, 72, and 81). However, by definition, endophytes are inhabitants of the plant endosphere and cannot be isolated from a soil. If the authors mean that the soil is a possible source of bacterial endophytes and because of that, they had collected soil of different management type for growing plants in a greenhouse - they should state it clearly.
A: We added a paragraph in the introduction where we specified that, except for seed-endophytes, the majority of the bacterial endophytes originate from the rhizospheric soil and enter into the plant tissues through primary and lateral root cracks, and through the tissue wounds occurring as a result of plant development. So that, the soil is a sink for bacteria able to efficiently colonize the root surface and also penetrate in to the plant tissue.
In Material and Methods, it should be described how the seeds for plant growing were collected.
A: The seeds were bought in a garden shop.
Lines 148-151. The description of medium content is not clear because the concentration of main medium components - DCP and TCP, are not specified.
A: line 172-174: the description of the medium was already fully reported. The DCP is not commercialized as molecule, but generated from the reaction between K2HPO4 and CaCl2·2H2O. The amount of TCP was already specified in the text (Ca3(PO4)2, 40 g l−1).
Lines 166-169. Once again, it should be specified, why namely this set of antibiotics was used. If it is a standard procedure to test the antibiotic resistance profile of a bacterial strain, with the standard set of antibiotics in standard concentrations - the relevant reference should be given.
A: The screening of antibiotic resistance/sensitivity is a fundamental step of the selection of biofertilizers. This step is usually performed before formulation and commercialization, in order to limit the spreading of antibiotic resistance in environment due to horizontal gene transfer. Using plant growth promoting bacteria that are resistant to antibiotic usually applied in human therapy is considered as a high risk for human health. The molecules considered in this work have been selected in relation to the type of microorganism we used (Pseudomonas); they are wide spectrum antibiotics or antibiotics having gram negative cells as their main target.
Lines 254 and 416. "Based on the taxonomic identification" - the role of taxonomic identification in the selection of bacterial endophytes for the analyses is unclear. Does it mean that the strains belonging to specific species/genera were selected? It does not follow from what is written in M&M and Results.
A: As stated, the strains have been selected according to their physiological activity and identification (possible plant and human pathogens have been obviously excluded). However, it is also true, that the literature produced by our scientific team is mainly focused on Pseudomonas.
Figure 6 – the question still remains: if 0 = absence of evident symptoms, what does -0.2 mean?
A: The point at -0.2 was fixed in order to highlight the black line of the control plants. Anyhow, Figure 6 has been modified as suggested.
Once again, both Results and Discussion are not divided into subsections (according to the physiological traits, for example), and this complicates reading and understanding the text.
A: According to the previous request of the reviewers, we separated the Results and Discussions sections, and we think the readability has been increased. However, looking at the formatting rules imposed by the journal there is no mention of the need to insert subtitles in Results or Discussion. Furthermore, we think that in introducing these subtitles the continuity of the discussion will be lost. We therefore prefer to maintain the current structure.
Once again, Discussion does not contain a sufficient analysis of the data presented. Except for the review of the previously published findings, it mostly just repeats the results obtained (as well as Conclusion). Thus, it is unclear for what purposes the authors used different plant species, grew them in the soil under different types of management, and isolated bacterial endophytes from different plant tissues. Just in order to increase the diversity of endophytes? It is OK. But why not to consider and analyze the influence of host plants (cucumber, sorghum, and tomato) and the soil where they were grown, as well as plant tissues (shoots, roots, leaves) on the endophyte composition and physiological abilities? For example, why namely the plants grown in the pasture soil gave the highest number of endophytes? Maybe because of the preferential soil physico-chemical properties?
A:The section conclusion was added to the original manuscript following the suggestions received at the first round of revision from reviewer 2. Reviewer 2 stated: “The paper must also contain the conclusion section summarizing most important findings and including the application perspectives”. We then wrote the conclusion following this opinion and reporting the main findings of this work (so that, it’s a repetition of the results) as well as the application perspectives (“The physiological activities of these bacteria are expressed directly inside the plants making them useful both as biofertilizers and biopesticides. Moreover, the broad host range showed by many bacterial endophytes makes them a powerful tool in the development of sustainable agriculture”).
The influence of the soil type and of the plant species on the endophytic community was already discussed (see line 420-424). We added another sentence (line 418-421) in order to explain the high amount of culturable endophytes isolated from pasture soil. As stated, on line 421-424, the influence of the host plant or of the soil should be studied not only on culturable bacteria but at the community level with molecular tools. This study was focused only on culturable bacterial endophytes that represent really a small portion of the rhizospheric/endophytic community. Therefore, making considerations or hypotheses about the influence of the plant or of the soil on a portion representing perhaps around 1% of the full bacterial community does not have any meaning.
In my opinion, for a research paper it is not enough just to answer the question "what", it should also explain "how" and "why". The manuscript in its current state does not contain such sufficient explanations.
Still, the language of the manuscript needs to be checked and corrected.
A: The language of this new version of the paper has been fully reviewed by one of our co-Authors who is a native English speaker.
All other corrections and suggestions are inserted into the attached PDF version of the manuscript.
A: All the suggestions present in the pdf file have been addressed.
It should be explained how rifampicin resistance is associated with true endophytism Line 306-307. “In order to assess whether the eight strains were the true endophytes, we selected spontaneous 265 rifampicin resistant mutants of each strain”
A: Rifampicin resistance is not associated with true endophytism. Simply, rifampicin resistance is a marker used to distinguish our strain from other environmental microorganisms that may contaminate the pots where the plant is growing. In fact, the plants were cultivated in a growth chamber that does not ensure sterile conditions (although the sands, the pots and the solution used to feed the plants have been sterilized).
Reviewer 3 Report
The second version of the presented manuscript is improved compared to the earlier version.
However I still think that the Figure 1 is not very informative and important for understanding of the scientific work described in this manuscript. Therefore it can be deleted or included in Supplementary.
Author Response
Response to Reviewer 3
The second version of the presented manuscript is improved compared to the earlier version.
However I still think that the Figure 1 is not very informative and important for understanding of the scientific work described in this manuscript. Therefore it can be deleted or included in Supplementary.
A: According to one of the requests of reviewer 2, we eliminated the coordinates of the three sites from the Materials and Methods section that are present on Figure 1. Therefore, we decided to maintain Figure 1 in the main text of the paper.
Round 3
Reviewer 2 Report
The manuscript looks better. The following concerns remained and arose:
Lines 57-62: According to the logic of Introduction, this new paragraph should be placed third, after the endophyte definition.
Still, it is unclear how the seeds were collected – in three regions in equal amount or in a different way?
Lines 187-188: Still, this sentence is appropriate for the Results section but not for M&M.
Table 1. "sp." should not be written in italic. "spp." means many species and cannot be used for a single strain.
Discussion and Conclusion. Still, the question – whether and how species of plant (cucumber, sorghum, and tomato) and kind of plant tissue (roots, shoots, leaves) affected the endophyte composition and physiological activities – remained unanswered (lines 420-424 only listed the plant species but do not contain any explanation).
In Conclusion, it is difficult to separate the data obtained in the current study from the previously published data.
And my major comment is the following. The authors must take into account that the journal where they want to publish their manuscript is not a specific bacteriological journal. Thus, the content, especially concerning the methodology used should be clearly understandable not only for bacteriologists, but also for a broad spectrum of readers. Because of that, the authors should clarified all aspects concerning the antibiotic resistance profile, the rifampicin resistance, as well as all abbreviations (for example, CFU), etc. not only in the respond to a reviewer, but namely in the text.
The attached PDF file contains other corrections and suggestions.

Author Response
The manuscript looks better. The following concerns remained and arose:
Lines 57-62: According to the logic of Introduction, this new paragraph should be placed third, after the endophyte definition.
A: The paragraph has been moved according to this request.
Still, it is unclear how the seeds were collected – in three regions in equal amount or in a different way?
A: As stated in the previous letter of revision, we bought the seeds in a shop garden. In the Materials and Methods, we added this information, between parenthesis, indicating the brand of the seeds. Then, as explained in the text, the seeds were surface sterilized and grown in the three different soils in order to isolate new endophytes.
Lines 187-188: Still, this sentence is appropriate for the Results section but not for M&M.
A: We moved the sentence according to this request to line 45-50
Table 1. "sp." should not be written in italic. "spp." means many species and cannot be used for a single strain.
A: These typos have been corrected.
Discussion and Conclusion. Still, the question – whether and how species of plant (cucumber, sorghum, and tomato) and kind of plant tissue (roots, shoots, leaves) affected the endophyte composition and physiological activities – remained unanswered (lines 420-424 only listed the plant species but do not contain any explanation).
A: As previously explained, this study was focused only on culturable bacterial endophytes. The reviewer should take into account that the amount of the culturable bacteria in soil has been estimated to 2%. Only a small proportion of these bacteria colonize the rhizosphere; and only a small proportion of the rhizospheric bacteria are able to colonize the interior of the plants. Therefore, making considerations or hypotheses about the influence of the plant or of the soil on a portion of the microbiota representing dramatically less than 2% of the full bacterial community does not have any meaning.
In Conclusion, it is difficult to separate the data obtained in the current study from the previously published data.
And my major comment is the following. The authors must take into account that the journal where they want to publish their manuscript is not a specific bacteriological journal. Thus, the content, especially concerning the methodology used should be clearly understandable not only for bacteriologists, but also for a broad spectrum of readers. Because of that, the authors should clarified all aspects concerning the antibiotic resistance profile, the rifampicin resistance, as well as all abbreviations (for example, CFU), etc. not only in the respond to a reviewer, but namely in the text.
A: We partially agree with the reviewer. The workflow of this paper is not simple, but it is also true that methods such as the MIC evaluation, the plate counting (determination of the CFU), as well as the antibiotic diffusion plate test are well described in every elementary microbiology textbook. We tried to be as clear as possible, the manuscript is now improved thanks to the reviewer’s suggestions, but we have to keep in mind that we are writing a scientific paper and not an academic textbook or laboratory manual.
Other points:
Figure 3 legend: The MIC method can be used to evaluate salt tolerance as well as the tolerance to antibiotics, heavy metals, organic pollutants and so on. This is the meaning of the sentence reported in the Figure legend 3 “test showing the tolerance of bacterial strains to specific molecule”.
The measurement of the MIC is done looking at the growth of the bacterial strain inside each single well of the microplate. So that, the sentence “The well was considered positive when the presence of a bacterial cell pellet was observed” is correct.
The reason for the choice of the antibiotics used in the determination of the antibiotic resistance sensitivity profile has been added in the text line 172-173.
The reason the strains have been tagged with spontaneous resistance to rifampicin has been explained on line 182-183.
Conclusion section: “On the contrary, the plant growth-promoting bacterial endophytes were found to limit the concentration of sodium in plant shoots through a still unknown mechanism”. This is a sentence referring to our strains, so a reference is not necessary.